# Artificial Intelligence (AI) in Brazilian Digital Journalism: Historical Context and Innovative Processes

**Moisés Costa Pinto** [1,2,*] **and Suzana Oliveira Barbosa** [1]

1. Online Journalism Research Group (GJOL), Federal University of Bahia, Salvador 40020-160, Brazil; suzana.barbosa@ufba.br
2. Communication and Design Department, Jorge Amado University, Salvador 41745-130, Brazil
* Correspondence: moisescosta@ufba.br

**Abstract:** This article investigates the historical uses and types of artificial intelligence (AI) systems and resources in Brazilian journalistic products. It is a work anchored in critically analyzing the literature on the subject, mapping and observing cases, seeking to identify uses and innovative processes, and analyzing AI projects for journalism. A search was carried out in web repositories, specifically Google, Google Scholar, and Scopus, using the terms: "inteligência artificial" + "jornalismo", "bot + jornalismo", "Geração de linguagem natural [NLG] + jornalismo", "aprendizado de máquina [machine learning] + jornalismo", and "algoritmos + jornalismo". The corpus analysis (N = 45) includes the evaluation of the impacts of AI on the production and distribution of news in the context of Brazilian digital journalism. We try to answer questions about the uses of databases, approximation with platforms, uses of shared codes, connections with other Ais, and sources of funding, and whether they are backend or frontend initiatives. In a parallel investigation, we try to identify if Brazilian newsrooms are officially using ChatGPT, a generative AI. The findings point to advances in using low-cost and low-impact AI, with the predominance of bots. The great availability of this kind of AI in web repositories is believed to facilitate native digital media to incorporate innovative processes in using these technologies.

**Keywords:** artificial intelligence (AI); digital journalism; automated journalism; innovation; Brazil; digital media



## 1. Introduction

As digital technologies advance in the real world, algorithms also advance in society. The global village, alluded to by McLuhan (1963), or our information society (Castells and Espanha 1999; Burch 2005; Lemos 2015), would not be possible without the logical power of algorithms and computers. An algorithm is a conceptual sequence of steps that must be expressed in any computer language, human language, or logic (Gillespie 2014). They perform (Lemos 2021) processes of mediatization, datafication, and the consequent abundance of information (Boczkowski 2021) that leads us to the era of big data (Kitchin 2014; Siegel 2013), where bits about the most diverse aspects of everyday life are processed, stored, treated, analyzed, etc. For Lemos (2021), datafication makes it possible to convert any and all actions into traceable digital data, producing diagnoses and inferences in the most diverse domains. This datafication is only possible thanks to digital technologies, which have algorithms at their core. They are also at the heart of big tech (Birch and Bronson 2022), the so-called high-tech platforms, which foster the platformization of society (Poell et al. 2020) and, in this context of data capitalism (Fonseca 2020; Sodré 2021), they generate participatory simulacra, helping to shape behavior and attitudes (Sodré 2021), often through predictions mined from "machine learning—computers automatically developing new knowledge and capabilities by feeding furiously on modern society's largest and most powerful unnatural resource: data" (Siegel 2013). Lemos (2021) calls this system "algorithmic intelligence," which should be thought of as a set of methods for

collecting, processing, and treating data to make predictions. These systems use data that they help create, causing new data to be produced continuously, according to Siegel (2013), retroactively feeding big data. So, data performs. Its performativity (Blouin 2020; Lemos 2021) refers to the idea that it does not simply represent reality but plays an active role in shaping and producing certain outcomes or subjectivities. This suggests that collecting, analyzing, and using data can have performative effects, influencing the behavior and decisions of individuals and institutions.

What has brought us to this point, where we are experiencing the predominance of data in our society, is the "algorithmic turn" (Linden 2018; Napoli 2014) enhanced by the advances in digital technologies. We can say that only algorithms and artificial intelligence (AI) can deal with the performativity of the data (Lemos 2021) that we continually produce every day.

Therefore, this paper addresses the intersection between artificial intelligence (AI) and contemporary digital journalism (Salaverría 2019; Steensen et al. 2021; Alves 2006; Perreault and Ferrucci 2020; Ferrari 2007)—both journalism and artificial intelligence are actors that are transforming society. More specifically, we look at artificial intelligence initiatives in Brazilian digital journalism. We will conduct a historical survey to investigate the uses and types of artificial intelligence (AI) systems and resources in Brazilian journalistic products. Some examples include bots, chatbots, machine learning, natural language generation, visualization, image creation, text generation, and story automation. These resources are used by various media, such as G1 and Grupo Globo, UOL, Folha de S. Paulo, Estadão, Núcleo Jornalismo, Aos Fatos, Serenata de Amor, and Jota. We will also answer questions about innovation, use of databases, approach to platforms, use of shared codes, connections with others AI, source of funding, and whether they are backend or frontend initiatives. In a parallel investigation, we will analyze whether Brazilian newsrooms officially use ChatGPT, a generative AI.

## 2. Artificial Intelligence (AI)

But what about artificial intelligence? Is it a machine that thinks, as Turing (1950) proposed? To begin with, algorithms are at the core of AI, as they are the foundation that makes any AI system work, like any digital system, whether they have virtual or physical interfaces.

> As the core of artificial intelligence, algorithms are widely used in news production and distribution by major media, and have had a significant impact on the field of news dissemination. In the field of news production, algorithms can help journalists find valuable clues and complete news reports, and provide matching news information to audiences in the field of news distribution.

(Zhang 2022)

Algorithms materialize the so-called smart media (Zhang 2022), while data are the key pieces that stand out in the high-tech journalism ecosystem (López-García and Vizoso 2021), whose scope is expanding more and more due to innovative digital resources, devices, and technologies. These range from drones, 360° video, immersive audio (binaural), virtual reality, augmented and extended reality, newsgames, holograms, 3D printers (Pavlik 2019; Silva 2022; Lima and Barbosa 2022), the use of machine learning, natural language processing (NLP), natural language generation (NLG)—which, for example, can automate the processes of producing, investigating, editing, publishing, distributing, and consuming journalistic content.

Artificial intelligence, in turn, is a technological tool that aims to mimic human actions and tasks (Cardozo et al. 2020) as they are, and, in journalism, contribute to (re)aggregating facts and data (structured or not) into information, and organizing and producing knowledge. In journalism, AI can be used for repetitive and "boring" tasks, such as systematizing financial reports and databases, as well as for searching for stories, writing, publishing,

disseminating, and interacting with audiences (Baldessar and Zandomênico 2022; Linden 2018; Diakopoulos 2019; Marconi 2020; Túñez-López et al. 2021).

Considering the forecasts (Kunova and Granger 2022; Newman 2023), the trend for this and the next few years is exponential growth and consolidation of the use of resources, tools, and technologies based on artificial intelligence (AI) by newsrooms, whether legacy media or digital natives.

The title phrase of Bruno Latour (1994)'s book "We Have Never Been Modern" can help problematize the advances of AI in journalism. Latour, in an analysis of the Anthropocene, points out that even in a society that is widely connected and poses as a global village (McLuhan 1963), there is a huge gap in the spread of all the technological achievements and resources of so-called modernity. Many still live on the fringes of cutting-edge technological development (Latour 1994) without access to digital technologies. According to the French author, technology does not arrive equally or simultaneously in every corner of the planet. This is also true for journalism, which mirrors the problems and contexts of each location. The journalism practiced in Brazil will not necessarily be done with the same bases, processes, and resources as in the Global North (Chan 2021).

Since adapting to new technologies and digital resources requires constant financial investment, not all news organizations, even the largest ones, can balance technological, editorial, and professional innovation and their management processes. This explains the slower pace of adoption of technologies associated with artificial intelligence. Decapitalization, aggravated by cyclical economic crises, points to a probable dependence on funding allocated through projects and partnerships with large platform companies (especially Google and Meta) to continue innovating, as is already happening in several countries. While platform companies dominate the field of AI, providing services, tools, and infrastructures, news organizations lack almost all resources and depend on big tech to conduct research and development (Simon 2022). However, in the results session, we will see data about the partnerships between vehicles and platforms that use AI in their initiatives.

## 3. AI and Journalism in Brazil

Previous research (Canavilhas and Giacomelli 2023; Cabral 2022; Cabral and Siqueira 2022; Essenfelder and Sant'Anna 2022; Essenfelder et al. 2019; Araújo 2017; Pinto 2021; Ioscote 2021; Dalben 2020; Carreira 2017; Paganotti 2020; Oliveira and Costa 2020; Träsel 2014) has addressed and exemplified the use of artificial intelligence in Brazilian journalism, pointing out trends, ethical issues, and innovations. In this paper, we conduct a systematic review of the cases pointed out, with an expansion and survey of the history of the penetration of AI that has been and is being officially used by digital native media, legacy media sites, fact-checking agencies, and experimental academic initiatives in Brazil. We also analyzed the impacts, types of AI, their spread, and innovations in news production in Brazilian digital journalism.

Lopezosa et al. (2023) say that "innovation" is the incorporation of new technologies, methodologies, and strategies into journalism. This encompasses exploring unexplored journalistic formats, including immersive journalism, 360° video reporting, virtual reality, and AI-automated journalism. In addition, innovation involves the assimilation of new technologies, actors, and practices to address the obstacles encountered by the creative industry (Bossio and Nelson 2021) in the face of changes in society, technology, the economy, and politics. On the other hand, innovation in journalism can also be understood from Franciscato's (2014) definition as adopting technological initiatives and processes that alter the media's business positioning, ways of working, and journalists' profiles. It can also be understood as the adoption of news products (formats and narratives) and modes of consumption. Moreover, this process of incorporating innovation into journalism, especially digital journalism in recent years, is ongoing (Barbosa 2014).

The adoption of artificial intelligence (AI) by news organizations is uneven (Lima Santos et al. 2022; Simon 2022), as it is still expensive and requires high investments, limiting its use, especially by those based in the Global South. This is the case in Brazil, where

studies indicate that the use of AI by national news organizations is still sporadic (Carreira 2017; Araújo 2017; Essenfelder and Sant'Anna 2022), although there are significant cases as reported in previous research (Pase and Pellanda 2019; Essenfelder et al. 2019; Dalben 2020, 2022; Santos 2020; Pérez-Seijo et al. 2023) and verified by us in the empirical research for this article.

The reason why new systems and tools based on generative artificial intelligence (GAI) are not used in every media outlet, especially in Brazil, also lies in the transnational differences in the distribution of technology, which is based, among other things, on the distribution of capital. In this scenario, Brazilian digital journalism finds itself in a marginal location in terms of development and access to technological and financial resources. Furthermore, as we shall see, this implies how and what types of initiatives to use AI are implemented in Brazil.

## 4. Methodology

From a methodological point of view, to critically study the presence of artificial intelligence (AI) in Brazilian digital journalism, we broadly mapped the synergy of these technologies in national media outlets. By observing the cases, we aimed to identify innovative uses and processes and answer questions about the impacts and effects of these journalistic AI initiatives and approaches to platforms. The research was carried out in two complementary stages:

(a)    an initial search between 1 May and 30 May 2021;
(b)    a complementary search between 1 April and 20 April 2023.

The second search was necessary to complement the first once new examples, including before 2021, emerged after the first search. Both searches took place in Google Scholar, Scopus web repositories, and Google's web search engine. First, we choose Google Scholar and Scopus for their enormous academic base, including theses and dissertations. In his turn, Google Web search was selected to fulfill the corpus with initiatives that have not yet been studied, and that are found only on websites and social media. In fact, the major initiatives that we analyzed had no place in academic studies yet, and this is one of the limitations of using academic repositories in this kind of research, whose purpose is to grab all possible uses in one particular period of time. Another limitation of the research is in the algorithms of these tools themselves, with biases in their codes written to meet companies' objectives, in what their search methodology shows research results that best meet what these companies believe to be the most viable considering their interests—this algorithmic methodology is a black box that we cannot access, we only have clues. This brings us to another critical point in this type of research: the need to use more than one web repository to obtain the best possible results—which we did.

We used the following terms, illustrated in Table 1.

**Table 1.** Search terms on artificial intelligence in Brazilian digital journalism.

| Search Terms |
|:---:|
| "inteligência artificial" + "jornalismo" + "Brasil" |
| "bot" + "jornalismo" + "Brasil" |
| "Geração de linguagem neural [NLG]" + "jornalismo" + "Brasil" |
| "aprendizado de máquina [machine learning]" + "jornalismo" + "Brasil" |
| "algoritmos" + "jornalismo" + "Brasil" |
| "jornalismo automatizado" + "Brasil" |

Source: Author.

We collected information from websites, blogs, social media posts, academic articles, dissertations, and theses that pointed to examples of artificial intelligence in Brazilian digital journalism. After this first stage, in order to compose the corpus, all the initiatives

found were filtered to determine whether they were automation initiatives using artificial intelligence in at least one stage of the journalistic production process—from a perspective of collaboration between humans and non-humans (Latour 1994, 2012; Lemos 2013). This answer was made possible by reading all the publicity material, documentation, and/or repositories—if available. Once confirmed that these were automation initiatives, further reading of these materials was undertaken for subsequent categorization with the help of an online Google Forms form, one of the most used online survey tools and useful in various academic activities (Mota 2019; Raju and Harinarayana 2016). We used the research tool for its practicality in creating databases—in an online spreadsheet—based on categorization by filling out an electronic form and offering quick visualization of the data collected in graphs. This helped speed up the categorization and a fast visual analysis, which included graphs assessing the type of AI, their technology, interactions, and relationships with platforms and understanding how these technologies position themselves as innovative.

However, the 2023 research update was also concerned with advancing generative artificial intelligence (GAI), especially ChatGPT. GAI are AI systems that can generate new data or content, such as images or text, based on patterns and examples from existing data sets (Corcoran et al. 2020) by connecting and interconnecting nodes and content nodes.

Pavlik (2023) believes that GAI could usher in an era of potential transformation of journalism and media content. This type of AI could help the journalism industry to identify more carefully the potential uses of this technology, which is currently underutilized compared to other industries (Santos 2020). Therefore, we conducted a specific search on Google and Google Scholar using the terms "chatgpt + jornalismo + Brasil" to identify the official use of ChatGPT in Brazilian newsrooms. This complementary research aligns with objectives at the point that we need to obtain a better approach to the uses of AI in Brazilian newsrooms as we understand that possible official uses of the tool by news companies can be addressed as initiatives in the research mains corpus. ChatGPT was chosen because it was the first text-based GAI in the chatbot format to hit the market, and it had relatively easy access with free and pro accounts—that have a relatively low cost (USD 20 per month), with facilitated access for Brazilian journalists. Other tools of the same type, such as Google's Bard, were not yet available in Brazil at the time of the research. In addition, Gondwe (2023) interviewed Sub-Saharan African journalists about their uses of ChatGPT and discovered that it has both positive and negative effects on journalism in developing countries. If there is a need to promote the responsible and ethical use of AI tools in journalism, it is also necessary to keep in mind that the chatbot consistently produces expected and stereotypical outputs, as:

> "[...] biases becomes apparent when the chatbot provides different responses to similar questions based on one's affiliations. Notably, ChatGPT tends to portray African countries and their leaders in a negative spotlight, with questions about them often concluding with anecdotes that connote poverty, disease, or corruption. (Gondwe 2023)

Studying how this is reproduced in the Brazilian context, similar to the African context, in the face of Global North power, but with its idiosyncrasies, is interesting for the overall context of our research. Nonetheless, the results of this research will be analyzed at the end of this article.

Finally, we used the nomenclature "initiatives" to refer to any and all appearances or uses of AI on the Brazilian journalistic websites that make up the sample. The term can better cover experimentation (López-García and Vizoso 2021) and how they can produce news with the help of "new" technologies, which until then had either been little used or not yet used in this task. Nevertheless, some initiatives are applied definitively, such as encompassing editorials. In contrast, others are undertaken on an ad hoc basis, such as producing a report that could only be carried out using AI to obtain and manage large volumes of data. In a scenario where there is still a literacy gap in the potential of AI in newsrooms, each initiative, whether it be the production of a report, the implementation of

a bot, or extensive use by an editor of a major media outlet, has similar impacts in terms of spreading the possibilities of their use in journalism.

## 5. Results

At the end of the searches, which began in 2021 and ended in 2023, we found 45 initiatives (N = 45), illustrated in Table 2, which applied or used artificial intelligence (AI) at some stage of the journalistic production process, from reporting, news, visualization, hyperinfographics (Cordeiro 2020; Silva 2022; scattering Harcup and O'neill 2017), curation, moderation, and interaction with the audience.

**Table 2.** Initiatives to use artificial intelligence in Brazilian digital journalism.

| Initiatives | Websites | Year | Types of AI |
|---|---|---|---|
| Attack Detector[1] | *Abraji* | 2022 | Scraping, statistical, analytical intelligence (metrics) |
| Robotox[2] | *Agência Pública, Repórter Brasil* | 2019 | Bot |
| Malu Bot[3] | *Agência Tatu* | 2018 | NLG/text production, scraping, data structuring, bot |
| Dandara[4] | *Agência Tatu* | 2022 | NLG/text production, bot |
| Fátima[5] | *Aos Fatos* | 2018 | NLG/text production, bot, chatbot/audience interaction, word processing, content curation |
| Radar[6] | *Aos Fatos* | 2019 | Scraping, data structuring, dashboard, visualization, statistics, content curation |
| Elas no Congresso[7] | *AzMina* | 2020 | Scraping, data structuring, dashboard, bot |
| Monitor de Discurso Político Misógino[8] | *AzMina* | 2021 | Scraping, data structuring, visualization, statistical |
| Robô Jornalista da Amazônia Azul[9] | *Bots do Bem* | 2021 | NLG/text production, scraping, bot, word processing, content curation |
| Da Mata Repórter[10] | *Bots do Bem* | 2021 | NLG/text production, scraping, bot, word processing, content curation |
| RadinhoReporter[11] | *Bots do Bem* | 2021 | NLG/text production, scraping, bot, visualization, content curation |
| TelinhaReporter[12] | *Bots do Bem* | 2021 | NLG/text production, scraping, bot, word processing, content curation |
| Busca Fatos[13] | *Busca Fatos* | 2022 | Bot, chatbot/audience interaction, content curation |
| Monitor—Newsletter Meio[14] | *Canal Meio* | 2016 | Scraping, content curation |
| Colaboranet[15] | *Colaboradados* | 2019 | Scraping, bot, word processing, cdwontent curation |
| Corona Repórter #COVID19[16] | *Corona Repórter* | 2020 | NLG/text production, scraping, bot, text processing, content curation |
| Rosa[17] | *Correio 24 Horas* | 2020 | Chatbot/audience interaction, content moderation/comments |
| Broadcast[18] | *Estadão* | 2010 | Scraping, data structuring, dashboard, visualization, text processing, statistics, content curation, analytical intelligence (metrics) |
| Basômetro[19] | *Estadão* | 2012 | Scraping, data structuring, dashboard, visualization, statistical |
| Análise de expressões faciais de candidatos em debate[20] | *Estadão* | 2018 | Image processing, analytical intelligence (metrics) |
| Estadão Infográficos[21] | *Estadão* | 2019 | Visualization, statistics |
| Maria Capitu[22] | *Estadão* | 2019 | NLG/text production, scraping, bot, visualization, image production, word processing, content curation |
| Estadão Dados[23] | *Estadão* | 2021 | Scraping, data structuring, visualization, statistician |
| Fake Check[24] | *Fake Check* | 2020 | Bot, chatbot/audience interaction, word processing |
| Folha Estatística[25] | *Folha de S. Paulo* | 2018 | NLG/text production, scraping, data structuring, visualization |
| GPS Eleitoral[26] | *Folha de S. Paulo* | 2018 | Scraping, data structuring, text processing, statistical, analytical intelligence (metrics), video processing |
| Acervo Folha[27] | *Folha de S. Paulo* | 2021 | Image processing |

**Table 2.** *Cont.*

| Initiatives | Websites | Year | Types of AI |
|---|---|---|---|
| Eleições G1[28] | *G1* | 2020 | NLG/text production, scraping, data structuring |
| Grupo Globo/Pixellot[29] [30] | *Grupo Globo* | 2019 | Dashboard, video production, statistics, content curation |
| Ruibot[31] | *Jota* | 2018 | Scraping, data structuring, bot |
| Beta[32] | *NOSSAS.ORG* | 2017 | Scraping, data structuring, bot, chatbot/audience interaction |
| Projeto Horus[33] | *Núcleo Jornalismo* | 2020 | Scraping, data structuring, dashboard, visualization |
| Weber[34] | *Núcleo Jornalismo* | 2021 | Bot |
| Monitor Nuclear/Political Pulse BR[35] | *Núcleo Jornalismo* | 2021 | Scraping, data structuring, dashboard, visualization |
| BedelBot[36] | *Núcleo Jornalismo* | 2022 | Scraping, data structuring, dashboard, visualization, statistician |
| BotPonto[37] | *Núcleo Jornalismo* | 2022 | Scraping, bot, video processing |
| Legisla Redes[38] | *Núcleo Jornalismo* | 2023 | Scraping, data structuring, bot, image production |
| Amplifica[39] | *Núcleo Jornalismo, AzMina* | 2022 | Scraping, data structuring, dashboard, statistics |
| Tramitabot[40] | *Radar Legislativo* | 2020 | Scraping, bot, chatbot/audience interaction |
| Ruralômetro[41] | *Repórter Brasil* | 2022 | Scraping, data structuring, dashboard, visualization, statistics |
| Mapa dos agrotóxicos na água[42] | *Repórter Brasil, Agência Pública* | 2019 | Scraping, data structuring, visualization |
| Rosie[43] | *Serenata de Amor* | 2016 | Scraping, data structuring, bot |
| Querido Diário[44] | *Serenata de Amor* | 2020 | Scraping, data structuring, dashboard, bot, visualization |
| Jarbas[45] | *Serenata de Amor* | 2020 | Scraping, data structuring, dashboard, visualization |
| Uol Eleições[46] | *Uol* | 2020 | NLG/text production, scraping, word processing, statistical |

Source: author.

Among the initiatives listed are 23 Brazilian media outlets and/or institutions (Figure 1). The startup Núcleo Jornalismo leads the number of initiatives, producing 15.55% (seven) of the sample. Estado de S. Paulo (Estadão) follows with 13.3% (six). The third-largest producer was the Bots do Bem project, which involves a partnership between the Federal University of Minas Gerais (UFMG) and the University of São Paulo (USP) and is, therefore, an experimental/academic initiative.

The initiatives were also categorized by the nature of their promoting medium/institution. Thus, it can be seen that the majority of them were conceived by digital native media, a total of 40% (18). Legacy media, on the other hand, account for 28% (13). We can also note the appearance of initiatives promoted by academic experiences, 13% (six). Fact-checking agencies accounted for 11.1% (five) of the sample. News agencies and non-governmental organizations (NGOs) accounted for 4.4% (two) each.

Regarding the timeline of these initiatives (Figure 2), which is important for understanding the evolution of adoption, the vast majority were launched from 2018 onwards, representing 13.3% of the total (six), with their peak occurring in 2020, at 22.22% (10). There was a break in the growth curve of initiatives of this type, most likely due to the COVID-19 pandemic, which began in 2020; there was a decrease in the growth curve in 2021 and 2022.

Of the 45 initiatives collected, 80% are still active. Only 20% (9) have been discontinued or are not updated. The majority of the sample was used to supply information, graphics, or automated summaries to the media outlets' websites, 66% (30). Specifically, 42% (19) are interactive web applications (chatbots, hyperinfographics (Cordeiro 2020), for example). X, in turn, was the third most used interface, comprising 37% of AI, especially bots. They also appear in newsletters (11%), Telegram (8%), WhatsApp (6%), Messenger (4%), apps (4%) and TV, print, Facebook, and Instagram (2% each).

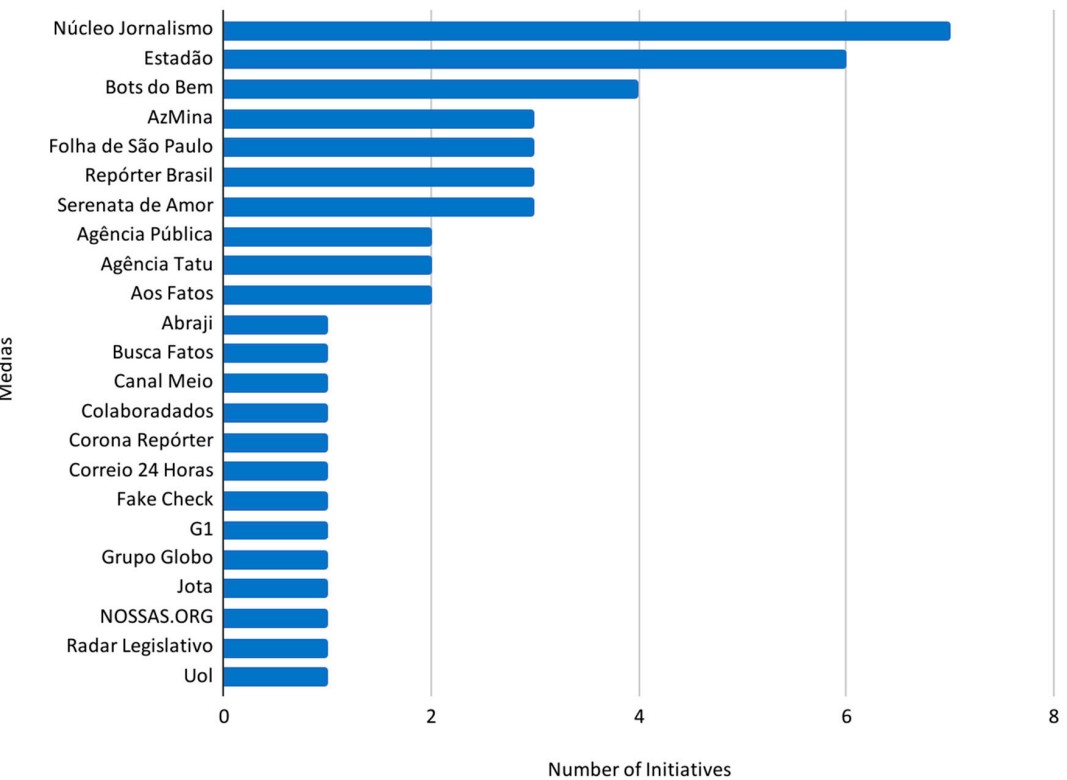

**Figure 1.** Number of initiatives by medium. Source: Authors.

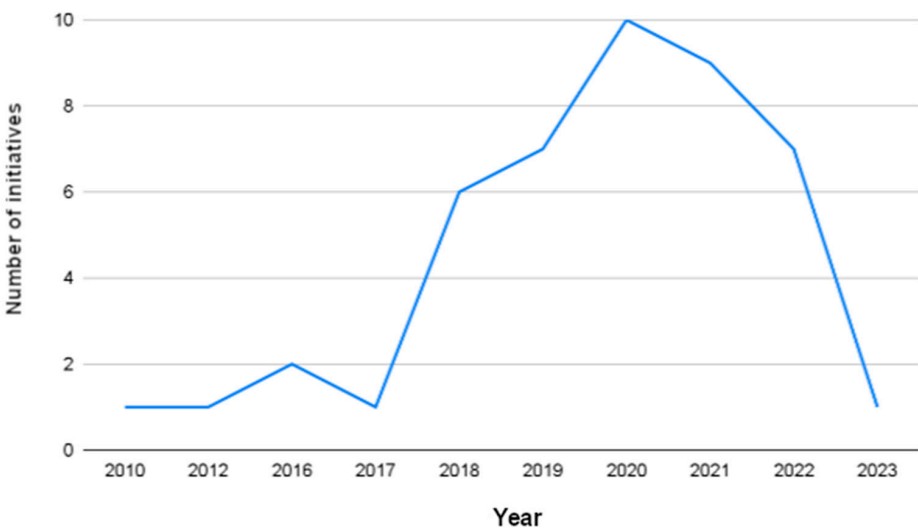

**Figure 2.** AI initiatives by year of launch. Source: Authors.

Regarding the technologies behind artificial intelligence, 75% (34) are based on algorithms scraped from databases (public and private), social networks, or the web. Bots and AI with data structuring account for 48% (22) each. Next is visualization AI with 35% (16) representation. AI aimed at statistical summarization and content curation is present in 28% (13) each; natural language generation (NLG) and dashboard creation are present in 26% (12) each. We also have word-processing tools, 24% (11), chatbots, 13% (six), and image production, 4% (two). Finally, video production, content moderation, and video processing complete the list with 2% (one) each.

Regarding the main objectives (Figure 3) for using AI in these initiatives, we have, in this order: data visualization, 35% (16), automated generation of statistics, 31% (14), and the creation of generative textual news (with NLG), 28% (13). This was followed

by infographics, 26% (12), push/notifications, 22% (10), political monitoring, 20% (9), and journalistic agenda generation and content curation, with 17% (8), each. Automated verification of fake news and disinformation, 13%, metrics and data insights, 8%, and audience interaction. Spreading, personalization, and environmental monitoring were 4% each. Judicial monitoring, video automation, financial market coverage, image processing, transmedia production, and video processing also appear in the sample, with 2% each.

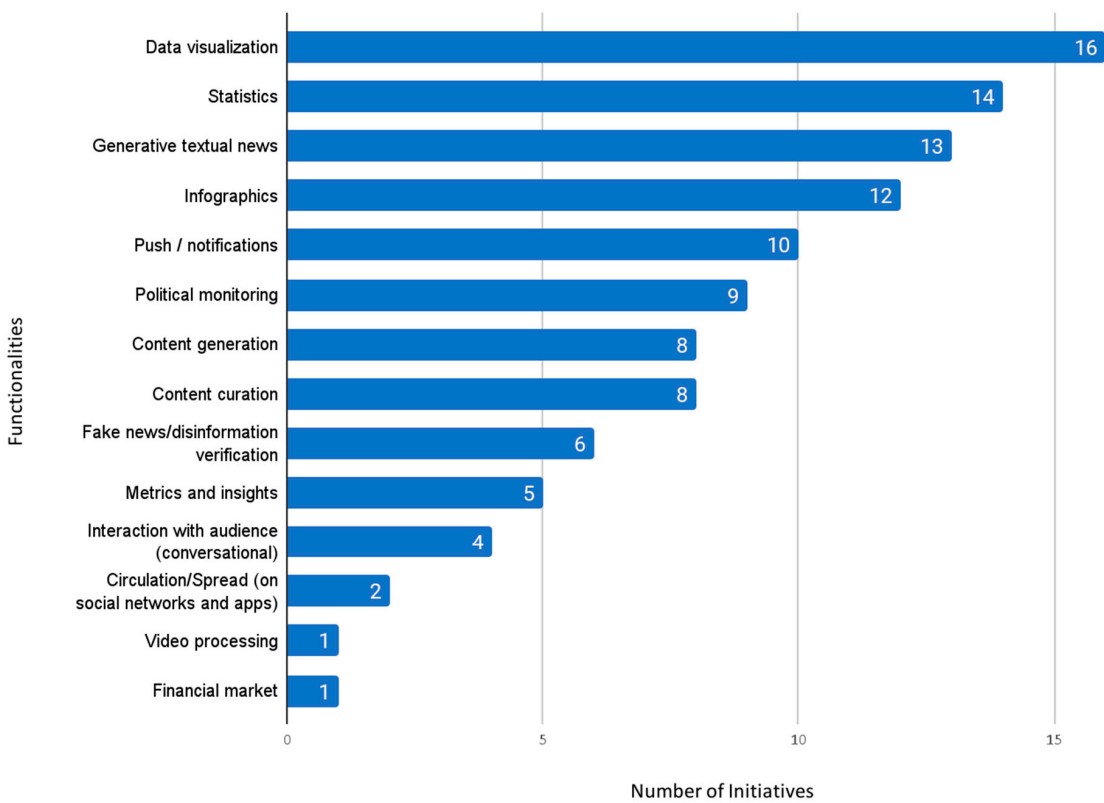

**Figure 3.** Objectives of the AI found. Source: author.

An interesting point is that the vast majority, 53% (24), of Brazilian initiatives cite open APIs, scripts, and codes, with licenses for unrestricted use without charge and available on the internet, often made available by previous initiatives. For example, some initiatives from Núcleo Jornalismo have codes created and made available by AzMinas. On the other hand, 17% (eight) do not use open-source codes, preferring their own or proprietary programming, and another 28% (13) do not inform or do not make it clear whether they use free codes. On the other hand, 60% (27) make their codes, APIs, and scripts available in web repositories, such as GitHub,[47] for other journalists to use in their automated journalistic projects. However, 22% (10) do not make their code available to third parties, while 17% (eight) do not report whether they make their programming available. However, it is important to note that so much availability can contribute to the spread of a broader culture of use and penetration of AI in Brazilian newsrooms, since it becomes much cheaper to produce initiatives from shared codes at no cost. This can create a culture and community of algorithmic literacy (Deuze and Beckett 2022) to spread automation technology in journalism.

When we analyze the connections between this artificial intelligence and other AI—whether used in journalism or not—it is possible to observe an integration gap: only 13% (six) of the corpus interact with other AI, such as Rosie and Jarbas, from the Serenata de Amor Project, who "talk to each other" in a process that complements their automated work. Even so, this integration is not clear in 17% (eight) of the initiatives, while the vast majority, 68% (31), have no connection reported or that could be observed in their documentation.

It is also interesting to note that the majority, 82% (37), of the AI that make up the corpus, do not use public data, such as government databases.

Most journalistic artificial intelligence (AI) still depends on some kind of supervision or editing by the human journalist in their content. Only 35% (16) publish or post automated content without the mediation of a human journalist/editor, whether on websites, messengers, newsletters, or social networks. In this respect, 33% have integrations with websites and/or web applications to enable automated publications or updates without direct human interference. AI that publishes directly to messengers without intervention represents 8% (four), and those that mediate newsletter content represents 4% (two).

As for integration with platforms, which are important for mediating journalism in contemporary times (Poell et al. 2020; Pereira 2022), we can see that X is the most used, with 58% (24) of the initiatives. Next comes Facebook, with 19% (eight), followed by Telegram, WhatsApp, Google, YouTube, and Instagram, with 7% each, and Microsoft and Spotify, with one integration each (2%). On the other hand, 14% (six) do not use any platform integration, which is unclear for another 19% (four). From this point, it is interesting to note that the vast majority, 68%, have no formal support from any platform. Even so, 13 initiatives have technological and financial support from platforms: Google, 15% (seven), Facebook, 6% (three), and Microsoft, WhatsApp, and YouTube with 2% (one) each.

In order to leverage the initiatives, the promoting media and institutions relied on a wide range of resources for financial sustainability (Figure 4). Most rely on funds from subscriptions, 53% (24), followed by crowdfunding, 40% (18), and advertisements, 31% (14). They also rely on funding from NGOs, 28%, platforms, 20% (9), governments, 13% (six), in the case of initiatives based on experiments at public universities, and direct investment, 2% (one). It is important to note that the vast majority rely on more than one funding source to support their viability (Maurício et al. 2017; Faria 2016)—a characteristic of the Brazilian journalism market. Notably, platforms such as Google, Meta, and Microsoft fund digital native initiatives. The funding projects of these big techs, roughly speaking, aim to encourage media and initiatives that bring innovation to the journalism market (Feil 2020; Jurno and d'Andréa 2020). It is a strategy aimed at developing cutting-edge innovations that can bring "news" to the field, but it is also a kind of "window shopping." In other words, it encourages projects that marketing can use to show how they—the platforms—can be "friendly" to journalism. The search for marketing stars is also part of the interest in implementing innovations, according to Dotzel and Faggian (2019).

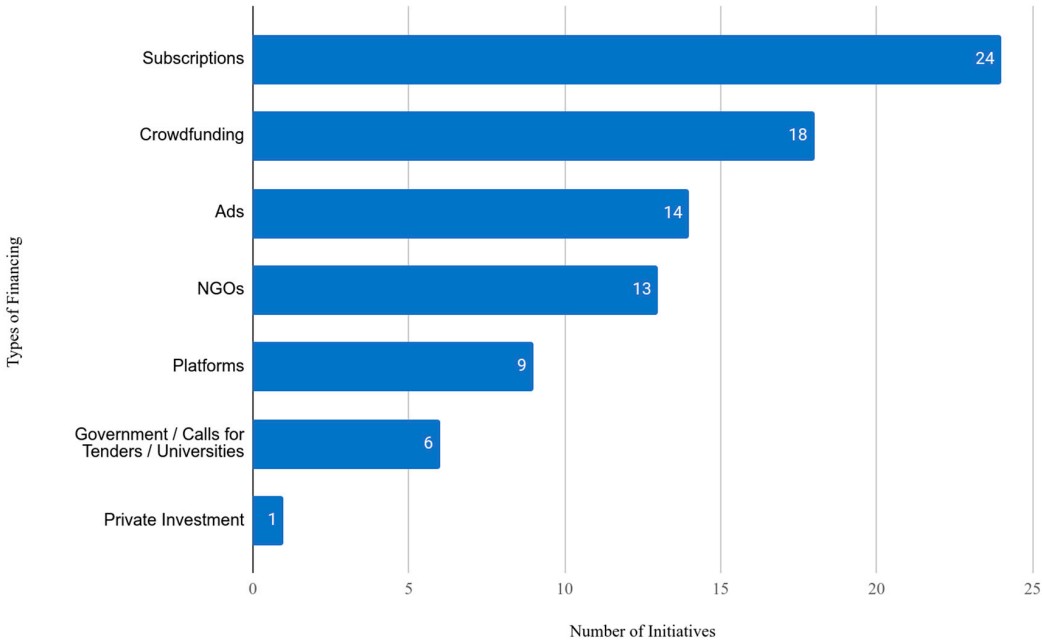

**Figure 4.** Website financing models. Source: Author.

However, only 35% (16) present or cite innovations in their documentation, presentation (of launches or reports published about them), or repository.

A final question analyzed dealt with how AI interacts with journalists and the public: audiences. We therefore methodologically divided the categorization into two sections:

(a) Frontend AI: In information science, a frontend program is one designed to be "seen" or interact with the end user (Abdullah and Zeki 2014; Pérez Ibarra et al. 2021); for us, bringing this concept to journalism and technology bases, a frontend AI applied to journalism is one that has direct contact, through the information it mediates or can interact with the audience, with or without human intervention, for example, the Rosie bot, which scrapes and structures data and publishes directly on X, where it can receive likes, replies, and comments from users.

(b) Backend AI: In information science, backend programming is programming that runs behind the visible interface, with access to data (Pérez Ibarra et al. 2021) and which is not seen or perceived by end users. To work with the backend, you need to know about databases, structures, and security aspects (*ibidem*) and how the logic of the technology used works. Translated into our research, a backend AI applied to journalism would not be available to the eyes of the audience but would be available for collaborations with other human actors (Latour 1994, 2012; Lemos 2013), journalists who would produce news in partnership with these AI—knowing, at the very least, the logic of the data and algorithms that make them work—for example, the generative artificial intelligence (GAI) that collaborated with the production of electoral news, based on data from the Superior Tribunal Eleitoral (Superior Electoral Court), for the G1 and UOL portals, on all of Brazil's more than five thousand municipalities. Thus, looking at the sample found, we can say that 77% (35) are AI with frontend interfaces, while 23% (10) focus only on the backend of journalistic production. Although they have limitations in operationalizing more acute interactions, many AI, such as bots and chatbots, open up new perspectives. They enable the audience to interact recursively, not only with the information but also with these non-human journalistic interfaces that produce news in collaboration or not with journalists.

## 6. Conclusions

One of the aims of this work is to help establish a historical framework for the application of AI in Brazilian digital journalism, especially given the progress made by generative artificial intelligence (GAI) at the end of 2022, such as with Midjourney and ChatGPT. Thus, this research can facilitate future comparative analyses regarding the use of AI in Brazilian digital journalism. However, one of the limits lies in the fact that the corpus was made up of AI communicated or reported by the Brazilian media—even those found in previous academic reports and studies. Thus, future research needs to extend into newsrooms to gauge the real extent and impact of the appropriation of AI by media and professionals, especially in the context of the spread of generative AI platforms.

When updating the survey in 2023, we expected significant advances in generative AI in newsrooms, such as ChatGPT. However, this was not the case. In a specific search, we noticed that journalistic organizations are still not using the OpenAI tool in an official and widespread way: we did not find any Brazilian organization that had adopted the tool and made this clear as of the closing date of this article. It is very likely that GAI applications scare professionals in the field, especially journalists, as they can be seen as "threats" to their already scarce jobs in newsrooms (Baldessar and Zandomênico 2022; Diakopoulos 2019; Marconi 2020). However, a shift induced by Generative AI (GAI) is on the way, as observed Pavlik (2023). The fast spread of GAI around our society, even with visible gaps between the global south and global north, can inevitably change how news media companies and journalists see and use it. Pavlik (2023) believes that AI tools such as ChatGPT could be used as an asset to assist a human journalist or media professional and, thereby, could be relevant to improving both the quality and efficiency of journalistic and media work.

It seems like generative AI is already a familiar technology to journalists or, at least, it is a technology that comes into the news world with a friendly interface to work with. The familiar web interface (a chatbot) and easy and cheap API integration to new tools can make technology literacy effortless for journalistic work, especially in underdeveloped countries. However, it is still necessary to amplify this research with surveys and interviews with Brazilian journalists who are using ChatGPT—even if their company does not allow it officially—to discover how it changes the way of daily work and the Brazilian journalist's perspectives with news produced.

If the media are not yet using ChatGPT and GAI platforms, the data we found, however, points to numerous initiatives aimed at solving small journalistic bottlenecks, such as, above all, accessing, managing, analyzing, and reporting from large databases. It means that are various opportunities to use AI in journalism, especially in data journalism.

The results show a profusion of AI spread across various journalistic sites and with diverse purposes (from generating stories to producing texts and graphics). However, the same data shows that this AI is being used as a tool to overcome analytical obstacles that human reporters would not be able to overcome or would take too long to handle and produce information. This is in line with Deuze and Beckett (2022) and Marconi (2020), who believe that AI applied to journalism, now and in the coming years, will play a much greater role in assisting with routine tasks that were previously very tiring, rather than replacing journalists entirely.

Another observable point in the corpus analyzed is that many AI are present in the Brazilian scenario as interactive tools on the journalistic frontend, such as bots, chatbots, and web applications. In the case of web applications, some are also in the context of hyperinfographics (Cordeiro 2020), where the user interacts with the information. Backend AI, such as those used by G1 and UOL to write election results, are not presented or have interfaces where users do not easily recognize their collaboration. This acknowledgment is only made later, in texts publicizing journalistic innovations by each outlet—of course, to publicize their achievements. Interestingly, the main backend AI analyzed is notably positioned in the big legacy media brands (G1, UOL, Estadão, Folha de S. Paulo, for example). Frontend AI, even with many limitations, is mostly implemented by digital natives (Serenata de Amor, Aos Fatos, AzMina, Jota, Núcleo Jornalismo, Agência Tatu, etc.). In fact, it shows the use of frontend AI for digital natives is more interactive, from the audience's perspective, and it can strengthen the relationship between automated news and the public. Ultimately, this could turn out to be an important tool for valorizing automated news by the public, which could facilitate the spread of this type of technology throughout newsrooms.

It is also fascinating to note that most frontend AI is based on open codes and shared by various projects, reinforcing the construction of a collaborative community for disseminating and consequent literacy of this AI by the Brazilian journalistic market in the context of digital natives. This community is responsible for the main front in the insertion of artificial intelligence into journalism in Brazil. As noted Lewis and Usher (2013), such a community can create an open-source journalism practice as they share technology with the purpose of providing a new framework that makes journalism more relevant in a participatory digital culture. It can also improve the transparency of the news production process.

Digital platforms such as Meta, Microsoft, X, and Google also play a key role in developing and disseminating AI in Brazilian digital journalism. They fund initiatives, mainly by digital natives, through their development programs—Google News Initiative, Meta Journalism Project, and Microsoft News Center—which look for potentially "innovative" initiatives to accelerate/develop. In a scenario of scarce resources, this platform funding becomes relevant for the media, especially digital natives, to launch initiatives that attract public attention and make their existence viable. On the other hand, news companies—especially digital natives—and journalists in Global South countries, such as Brazil, with a lake of resources, may see an even greater gap in relation to resources and professionals in the Global North, further increasing the dependence on big tech for access

to AI. This could create another technological gap between journalism practiced in the Global North and the Global South.

Additionally, in this scenario, the use of AI to initiate innovations seems to be a peaceful point: the limitations of contemporary digital journalism and the limited resources in Brazil mean that AI is a feasible alternative for producing information through data and even for producing large amounts of data (Attack Detector, for example) in order to summarize it into new information finally. The Brazilian example shows how news companies struggle to surpass boundaries of financial limitations to create their own solutions and, thus, be able to stay up to date regarding what technological innovation can contribute to improving processes and products using AI.

Moreover, noteworthy is how little publicity has been given to many AI projects. A few cases have gained a lot of repercussions, such as G1, which has been featured in the press and also in academic research, while the vast majority have hardly been reported within their media, such as UOL, which, like G1, used a generative AI to produce news based on electoral data for each Brazilian municipality in the 2022 general elections. Media outlets seem reluctant to use this technology and disclose that they are using it. It appears that they are afraid of how the public can react to news produced by AI, such as noted previously Noain-Sánchez (2022), as it is a technology that is not well known. However, the exact reasons in the Brazilian context still need to be explored in future research, ideally through applications of surveys, newsmaking observations, and interviews with the professionals involved in these initiatives to better capture what they think about it and the real impact of news AI initiatives in the market.

**Author Contributions:** Conceptualization, M.C.P. and S.O.B.; methodology, M.C.P. and S.O.B.; software, M.C.P.; validation, M.C.P. and S.O.B.; formal analysis, M.C.P. and S.O.B.; investigation, M.C.P., S.O.B.; resources, S.O.B.; data curation, M.C.P.; writing—original draft preparation, M.C.P. and S.O.B.; writing—review and editing, M.C.P. and S.O.B.; visualization, M.C.P.; supervision, S.O.B.; project administration, S.O.B.; funding acquisition, S.O.B. All authors have read and agreed to the published version of the manuscript.

**Funding:** This study was financed in part by the Coordenação de Aperfeiçoamento de Pessoal de Nível Superior—Brasil (CAPES)—Finance Code 001.

**Data Availability Statement:** The data used for this work can be made available to the public subject to data terms and conditions.

**Conflicts of Interest:** The authors declare no conflicts of interest.

## Notes

1    https://github.com/JournalismAI/attackdetector (accessed on 20 April 2023).
2    https://twitter.com/orobotox (accessed on 20 April 2023).
3    https://github.com/agenciatatu/malu (accessed on 20 April 2023).
4    https://www.agenciatatu.com.br/dandara/ (accessed on 20 April 2023).
5    https://www.aosfatos.org/fatima/ (accessed on 20 April 2023).
6    https://www.aosfatos.org/metodologia-radar-aos-fatos/ (accessed on 20 April 2023).
7    https://twitter.com/elasnocongresso (accessed on 20 April 2023).
8    https://github.com/fer-aguirre/pmdm (accessed on 20 April 2023).
9    https://twitter.com/BLAB_Reporter (accessed on 20 April 2023).
10    https://twitter.com/DaMataReporter (accessed on 20 April 2023).
11    https://twitter.com/RadinhoReporter (accessed on 20 April 2023).
12    https://twitter.com/TelinhaReporter (accessed on 20 April 2023).
13    https://buscafatos.com.br/ (accessed on 20 April 2023).
14    https://monitor.canalmeio.com.br/login/?next=/novidades (accessed on 20 April 2023).
15    https://twitter.com/colabora_bot (accessed on 20 April 2023).
16    https://twitter.com/CoronaReporter (accessed on 20 April 2023).

17    https://www.correio24horas.com.br/noticia/nid/correio-lanca-robo-que-permite-ao-leitor-sugerir-pautas/ (accessed on 20 April 2023).

18    https://www.estadao.com.br/economia/broadcast-lanca-novo-aplicativo-que-se-adapta-a-cada-usuario/ (accessed on 20 April 2023).

19    https://arte.estadao.com.br/politica/basometro/ (accessed on 20 April 2023).

20    https://www.estadao.com.br/infograficos/politica,o-que-revela-uma-analise-das-emocoes-dos-candidatos-durante-o-debate,923037 (accessed on 20 April 2023).

21    https://www.estadao.com.br/infograficos (accessed on 20 April 2023).

22    https://twitter.com/mariacapitu (accessed on 20 April 2023).

23    https://www.estadao.com.br/infograficos/saude,tratamento-precoce-perde-nas-redes-para-cpi-da-covid-e-vacina,1181660 (accessed on 20 April 2023).

24    http://nilc-fakenews.herokuapp.com/ (accessed on 20 April 2023).

25    https://www1.folha.uol.com.br/educacao/2018/04/estudo-inedito-indica-alta-chance-de-fraude-em-mil-provas-do-enem.shtml (accessed on 20 April 2023).

26    https://www1.folha.uol.com.br/poder/2018/10/veja-o-que-os-candidatos-mais-falaram-na-campanha-presidencial-e-em-sp.shtml (accessed on 20 April 2023).

27    https://www1.folha.uol.com.br/folha-100-anos/2021/11/video-mostra-trabalho-de-indexacao-de-25-milhoes-de-fotos-da-folha.shtml (accessed on 20 April 2023).

28    https://g1.globo.com/politica/eleicoes/2020/noticia/2020/11/12/em-iniciativa-inedita-g1-publica-textos-com-resultado-da-eleicao-em-cada-uma-das-5568-cidades-do-brasil-com-auxilio-de-inteligencia-artificial.ghtml (accessed on 20 April 2023).

29    https://oglobo.globo.com/economia/grupo-globo-aposta-na-inteligencia-artificial-em-transmissao-esportiva-1-24116586?utm_source=pocket_reader (accessed on 20 April 2023).

30    Partially acquired by Grupo Globo in a R$13 million investment.

31    https://twitter.com/ruibarbot (accessed on 20 April 2023).

32    https://www.beta.org.br/ (accessed on 20 April 2023).

33    https://nucleo.jor.br/projeto-horus/ (accessed on 20 April 2023).

34    https://twitter.com/weber_bot (accessed on 20 April 2023).

35    https://www.nucleo.jor.br/monitor/ (accessed on 20 April 2023).

36    https://nucleo.jor.br/bedelbot/ (accessed on 20 April 2023).

37    https://nucleo.jor.br/botponto/ (accessed on 20 April 2023).

38    https://twitter.com/legislaredes (accessed on 20 April 2023).

39    https://www.nucleo.jor.br/amplifica-form/ (accessed on 20 April 2023).

40    https://radarlegislativo.org/sobre/ (accessed on 20 April 2023).

41    https://ruralometro2022.reporterbrasil.org.br/ (accessed on 20 April 2023).

42    https://portrasdoalimento.info/agrotoxico-na-agua/# (accessed on 20 April 2023).

43    https://twitter.com/RosieDaSerenata (accessed on 20 April 2023).

44    https://queridodiario.ok.org.br/ (accessed on 20 April 2023).

45    https://jarbas.serenata.ai/dashboard/ (accessed on 20 April 2023).

46    https://economia.uol.com.br/noticias/redacao/2022/11/01/uol-publica-textos-de-resultados-das-eleicoes-para-mais-de-5500-cidades.htm (accessed on 20 April 2023).

47    https://github.com/ (accessed on 20 April 2023).

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
