# Peer review of "Artificial Intelligence (AI) in Brazilian Digital Journalism: Historical Context and Innovative Processes"

_journalmedia, doi:10.3390/journalmedia5010022_

Round 1

Reviewer 1 Report

Comments and Suggestions for Authors

I thank the authors for their contributions.

The authors perform a descriptive analysis of AI initiatives in Brazilian newsrooms. The method of data collection and analysis is straightforward and appropriate. The literature review encompasses significant literature on the topic.

The issue here is there is very little theoretical contribution to this submission. What does the manner of adoption of this technology by Brazilian newsrooms, and in the particular way that they are doing, teach us about how this technology can be adopted elsewhere, or in the future in Brazil and elsewhere? How relevant would a shift to Generative AI or ChatGPT be in that regard? What connections can we make between the findings of this study and other research aside from Deuze and Beckett (2022) and Marconi (2021)? How can it be improved upon by future research? In other words, what can this study teach us about AI and digital journalism outside the scope of the study itself? I believe these and other questions can be explored further in the conclusion and discussion.

Again, I thank the authors for their contributions and look forward to their future contributions.

Author Response

Dear. Thank you for your considerations. We made some adjustments to the final considerations to expand connections with other authors. Furthermore, we expand the considerations about ChatGPT, in addition to exploring its bibliographical contribution.

Reviewer 2 Report

Comments and Suggestions for Authors

Dear Author(s),

I have had the opportunity to thoroughly review your manuscript titled " Artificial Intelligence (AI) in Brazilian digital journalism:

historical context and innovative processes." I would like to commend you for the meticulous work you have done in crafting a well-grounded and structured study on the intersection of artificial intelligence (AI) and Brazilian digital journalism. Your introduction provides a solid foundation by effectively linking the advancement of digital technologies to the logical power of algorithms and computers, setting the stage for a comprehensive exploration of AI in journalism. The terminology used, such as "algorithmic intelligence" and "algorithmic turn," is adequately explained, contributing to reader understanding. Additionally, your efforts in providing clear definitions of terms like datafication and performativity significantly enhance the clarity of concepts. The outlined research questions regarding the uses and types of AI systems and resources in Brazilian journalistic products are commendably outlined.

However, I would like to offer some suggestions, particularly focusing on the methodology section:

  1. While the time frame for the initial and complementary searches is clearly defined, providing more explanation about the rationale behind the two searches and the factors influencing the decision for a follow-up search would add depth to your methodological approach.
  2. The choice of search terms and the selection of Google Scholar and Scopus web repositories, as well as Google's web search engine, are comprehensive. Adding a brief explanation of the rationale for selecting these sources, along with addressing potential limitations or biases associated with them, would enhance transparency.
  3. The utilization of Google Forms for categorization and analysis is practical and efficient. A brief justification for choosing this tool and highlighting its advantages in facilitating the creation of databases based on categorization would provide valuable context.
  4. Regarding the focus on ChatGPT in 2023, while the rationale is explained, providing clarification on how the introduction of ChatGPT in the Brazilian context aligns with the overall research objectives would strengthen this section. Additionally, drawing insights from other studies in the global south, such as the one conducted in sub-Saharan Africa (https://www.degruyter.com/document/doi/10.1515/omgc-2023-0023/html?lang=en), could enrich your discussion.

Finally, it would be immensely beneficial for the authors to address the "so what question" in their discussion of the findings. Providing a clear explanation of the meaning or implications of your findings would enhance the overall impact of your study.

Thank you for your dedication to this research, and I look forward to seeing the refinement of your manuscript.

Author Response

Dear. Thank you for your considerations. We made some adjustments to the methodology according to your suggestions. We also responded to your suggestion to expand the final considerations when exploring possible developments of the research.

Round 2

Reviewer 1 Report

Comments and Suggestions for Authors

I thank the authors for the inclusion of more details and implications of their research contribution.